# Survival of Women with Advanced Stage Cervical Cancer: Neo-Adjuvant Chemotherapy Followed by Radiotherapy and Hyperthermia versus Chemoradiotherapy

**DOI:** 10.3390/cancers16030635

**Published:** 2024-02-01

**Authors:** Jonathan Servayge, Ester P. Olthof, Constantijne H. Mom, Maaike A. van der Aa, Hans H. B. Wenzel, Jacobus van der Velden, Remi A. Nout, Ingrid A. Boere, Helena C. van Doorn, Heleen J. van Beekhuizen

**Affiliations:** 1Department of Gynecologic Oncology, Erasmus MC Cancer Institute, University Medical Centre Rotterdam, 3000 CA Rotterdam, The Netherlands; 2Department of Gynecologic Oncology, Amsterdam University Medical Centre, Centre for Gynecologic Oncology Amsterdam (CGOA), 1066 CX Amsterdam, The Netherlands; 3Department of Research and Development, Netherlands Comprehensive Cancer Organisation (IKNL), 3501 DB Utrecht, The Netherlands; 4Department of Radiotherapy, Erasmus MC Cancer Institute, University Medical Centre Rotterdam, 3000 CA Rotterdam, The Netherlands; 5Department of Medical Oncology, Erasmus MC Cancer Institute, University Medical Centre Rotterdam, 3000 CA Rotterdam, The Netherlands

**Keywords:** locally advanced cervical cancer, overall survival, disease-free survival, toxicity, chemoradiotherapy, lymph node debulking, hyperthermia

## Abstract

**Simple Summary:**

Women with locally advanced cervical cancer and nodal involvement remain a prognostically unfavourable group. Concurrent chemoradiation is considered standard treatment; however, alternative treatments have been investigated. Our main objective was to investigate overall survival and disease-free survival in triple therapy in locally advanced cervical cancer with nodal involvement. Furthermore, we wanted to compare triple therapy to standard chemoradiotherapy in a patient cohort with the same inclusion criteria. We included women with a tumour size of ≥6 cm, and/or pelvic lymph node metastasis of ≥2 cm and/or para-aortic lymph node metastasis of ≥1 cm. In our cohort of 370 patients, toxicity and survival of triple therapy is similar to chemoradiation with or without prior lymph node debulking. These findings suggest a role for hyperthermia in the management of locally advanced cervical cancer and could offer patients with nodal involvement an alternative treatment option.

**Abstract:**

Aim: To investigate and compare overall survival (OS), disease-free survival (DFS) and toxicity of women who underwent either chemoradiotherapy with or without prior lymph node debulking or upfront chemotherapy followed by radiotherapy and hyperthermia (triple therapy) for locally advanced cervical cancer (LACC) to identify a potential role for triple therapy. Methods: Women with histologically proven LACC and with International Federation of Gynecology and Obstetrics (FIGO) 2009 stage IB2 and IIA2 to IVA were included. Cox regression analyses were used for calculating hazard ratios and to adjust for confounding variables. A multivariable logistic regression analysis was used to examine the influence of covariates on toxicity. Results: A total of 370 patients were included of whom 58% (*n* = 213) received chemoradiotherapy (CRT), 18% (*n* = 66) received node-debulking followed by chemoradiotherapy (LND-CRT) and 25% (*n* = 91) received triple therapy (TT). Five-year OS was comparable between the three treatment groups, with 53% (95% confidence interval 46–59%) in the CRT group, 45% (33–56%) in the LND-CRT group and 53% (40–64%) in the TT group (*p* = 0.472). In the adjusted analysis, 5-year OS and DFS were comparable between the three treatment groups. No chemotherapy-related differences in toxicity were observed. Conclusion: This study suggests that the toxicity and survival of TT is similar to CRT or LND-CRT.

## 1. Introduction

Cervical cancer is the fourth most frequently diagnosed cancer and was the fourth most common cause of cancer death in women worldwide in 2020 [1]. In patients with LACC, i.e., FIGO 2009 stage IB2 and IIA2 to IVA, concurrent CRT is considered the standard treatment [2]. This would encompass external beam radiotherapy and brachytherapy, with platin-based chemotherapy. Extended-field radiotherapy (EFRT) and boost radiotherapy can be added depending on the presence and extent of lymph node metastases. In 2018, the FIGO revised the staging system of cervical cancer by adding stage IIIC1 and stage IIIC2, respectively for pelvic lymph node (PELNM) and para-aortic lymph node metastasis (PAOLNM), reflecting the significance of lymph node metastases as a prognostic factor.

Since the outcome in women with extensive nodal disease is poor [3], alternative treatments have been suggested. Adjuvant chemotherapy so far has not improved outcomes in unselected patients [4]. An alternative treatment strategy is neoadjuvant chemotherapy (NACT) followed by CRT. Recently, the INTERLACE trial reported that NACT followed by CRT significantly improves OS and DFS in LACC [5]. Two trials in which NACT followed by surgery was compared to CRT in LACC did not show improved OS or disease-free survival [6,7]. In patients with bulky lymph nodes, lymph node dissection or lymph node debulking (LND) may be performed prior to CRT in patients with LACC and PELNM or PAOLNM. A recent retrospective study found no difference in survival of LND in patients with LACC and suspicious bulky nodes [8].

A fourth alternative strategy in patients with LACC is TT, which consists of NACT, followed by radiotherapy (RT) and concurrent deep hyperthermia (HT). In TT, NACT aims to reduce the tumour volume, whereas HT is used as a radiosensitizer. HT increases the cytotoxic effects of ionizing radiation on cancer cells by local heating of the tumour up to 42 °C during 60 min exposure [9,10,11]. The value of HT in the management of locally advanced tumours was first illustrated by the Dutch Deep Hyperthermia trial [12]. In this randomised controlled trial (RCT) RT with HT was compared to RT alone in patients with advanced bladder, cervical and rectal tumours. In cervical cancer specifically, complete response was achieved in 83% of patients receiving RT with HT compared to 57% receiving RT alone. Furthermore, 3-year OS rates were 27% and 51% following RT and RT with HT, respectively. There has been one phase III RCT, the RADCHOC trial, that aimed to establish if RT with HT should be preferred over CRT in LACC [13]. The trial was closed prematurely because of poor accrual, but limited data (84 patients) suggested comparable outcomes for CRT versus RT with HT. A meta-analysis of two prospective studies by Yea et al. concluded that CRT with HT significantly improved OS in LACC patients without increasing acute and chronic toxicity [14].

In the current study we evaluated overall survival, disease-free survival and treatment-related toxicities in patients with locally advanced cervical cancer treated with chemoradiotherapy with or without prior lymph node debulking or upfront chemotherapy followed by radiotherapy and hyperthermia.

## 2. Materials and Methods

### 2.1. Patient Selection and Study Design

We included women diagnosed between 2009 and 2017 with FIGO 2009 stage IB2 and IIA2 to IVA squamous cell carcinoma, adenocarcinoma or adenosquamous carcinoma of the cervix. Women with a tumour size of ≥6 cm, and/or PELNM of ≥2 cm and/or PAOLNM of ≥1 cm on radiological imaging (short axis diameter) were eligible for inclusion. Patients treated with NACT followed by CRT, a history of pelvic radiation or FIGO 2009 stage IVB were excluded from this study. Common iliac lymph nodes were considered pelvic lymph nodes. Data on treatment and outcome of the patients in the CRT and LND-CRT group (treated in any Dutch hospital) were collected from the Dutch Cancer Society project ‘Chasing nodes, saving lives’ from the Netherlands Cancer Registry. From the Erasmus MC Hyperthermia database, we collected data on women treated with TT. In the Amsterdam MC, Amsterdam, The Netherlands, LND-CRT was performed; in the Erasmus MC, Rotterdam, The Netherlands, TT was offered and elsewhere in the Netherlands, the treatment consisted of CRT. The study was approved by the Medical Ethics Review Committee of Erasmus MC, Rotterdam, The Netherlands (MEC-2022-0246) and the Privacy Review Board of the Netherlands Cancer Registry (K22.018), The Netherlands.

Baseline patient and tumour-related characteristics, details on treatment and toxicity and survival data were retrieved from the two databases. All patients treated with triple therapy were assessed through CT-thorax-abdomen to detect the presence of lymphadenopathy and distant metastases, and an MRI to measure tumor size and local tumor growth. For patients treated with chemoradiotherapy or lymphadenectomy followed by chemoradiotherapy, 231 (82%) were assessed with PET/CT, 244 (88%) were assessed using MRI and 120 (43%) were assessed by CT scan. Bulky nodes were defined as PELNM ≥ 2 cm or PAOLNM ≥ 1 cm. The short-axis diameter of the largest suspicious node on radiological appearance was used for classification and analyses. Peri- and postoperative complications ≤ 30 days from surgery were scored by the Clavien—Dindo scale (grade 2 or higher) [15]. Chemotherapy-related toxicity ≤ 6 months after starting treatment was scored by the Common Terminology Criteria for Adverse Events version (CTCAE v4.03) [16]. CTCAE grade 3 or higher was considered relevant. For recurrent disease, the site of recurrence (central pelvic, pelvic side wall, para-aortic, abdominal wall, intra-abdominal and/or distant) and interval between treatment initiation and recurrence or death was defined. Vital status was checked by linkage with the Personal Records Database (censored at 31 January 2023).

### 2.2. Chemoradiotherapy with or without Nodal Debulking

Patients were treated according to national and international guidelines with RT (total dose 45–50 Gy), with concurrent single-agent platinum-based chemotherapy (cisplatin weekly 40 mg/m^2^), and brachytherapy until a minimal prescribed physical dose equivalent of 80 Gy (EQD2). EFRT was indicated if common iliac or para-aortic regions were involved. Individual details on radiotherapy doses were not registered in the database as part of this study. In the LND-CRT group, node debulking was performed prior to CRT. All patients treated with LND-CRT were treated in the Amsterdam MC. Lymph nodes >15 mm were debulked to increase local control and the chance of complete sterilization and subsequently, to reduce the radiotherapy dose and its associated toxicity. Histopathological review was conducted and data on extent of surgery (pelvic/para-aortic LND) were collected.

### 2.3. Triple Therapy

Patients were treated with platinum-based chemotherapy in a weekly (dose dense) schedule, namely cisplatin 70 mg/m^2^ with paclitaxel 90 mg/m^2^ on days 1, 8, 15, 29, 36 and 42, or carboplatin AUC4 + paclitaxel 90 mg/m^2^ on days 1, 8, 15, 29, 36 and 42. Patients started with cisplatin, however, mainly due to renal toxicity, they switched to carboplatin. Patients started combined treatment consisting of external beam radiotherapy, hyperthermia and brachytherapy two-to-six weeks after completion of chemotherapy. EFRT was indicated if common iliac or para-aortic regions were involved. During RT, patients were scheduled for five weekly hyperthermia sessions. The BSD-2000 3D system (BSD Medical Systems, Salt Lake City, UT, USA) was used for all treatments. Bowman probes were placed intraluminally in the bladder, vagina and rectum with closed-tip catheters for thermometry. Every 5 min thermal mapping along the catheters was performed with a step size of 1 cm and a maximum map length of 14 cm. Pulse rate and blood pressure were automatically measured before and every 5 min during treatment. Oral temperature was measured at 0, 15, 30, 60 and 90 min. At a power output of 400 W at 77 MHz heating was started. Patients were carefully instructed to report any discomfort due to too high temperatures in normal tissue during treatment. Treatment settings for power, phase and frequency were adjusted accordingly if symptoms developed. If no symptoms developed, 100 W was added to the power output every 5 min. The treatment objective was to achieve intraluminal temperatures of 40–43 °C as homogeneously as possible. For all patients, 90-min sessions were scheduled for each hyperthermia treatment consisting of 30 min of heating up and 60 min of actual treatment time. Data on the temperature factor or thermal dose were not available on a patient level.

### 2.4. Statistical Analysis

Differences of baseline characteristics between treatment groups were investigated. Continuous variables were compared using the one-way ANOVA or Kruskal—Wallis test and categorical variables using a chi-square test. To estimate OS and DFS rates, the Kaplan—Meier method was used. OS was defined as interval between date of diagnosis and death of any cause or censoring (e.g., emigration). DFS was defined as the interval from treatment initiation until death of any cause, or recurrence (whichever occurs first). Cox regression analyses were used for calculating hazard ratios (HRs), with 95% confidence intervals (CI). The proportional hazards assumption was tested by plotting scaled Schoenfeld residuals and conditions were met. Analyses were adjusted for: age, FIGO 2009 stage (<stage III or ≥stage III), histological type (squamous or non-squamous), tumour size in mm and location of lymph nodes (PELNM and PAOLNM). A multivariable logistic regression analysis was used to examine the influence of covariates on toxicity (i.e., chemotherapy related or postoperative complications). A *p*-value of less than 0.05 was considered significant. All analyses were performed in Statistical Package for the Social Sciences (SPSS) version 28.0.1.0 (IBM Corp. Released 2021. IBM SPSS Statistics for Windows, Version 28.0. Armonk, NY, USA) or Statistical software for data science (STATA) SE 17 (StataCorp, College Station, TX, USA) softwares.

## 3. Results

### 3.1. Baseline Characteristics

In this study, 370 patients met the inclusion criteria of whom 58% (*n* = 213) received CRT, 18% (*n* = 66) received LND-CRT and 25% (*n* = 91) underwent TT. Patient, tumour and treatment characteristics are summarized in Table 1. Median age was 50 years in the CRT and LND-CRT group, range 22–86 and 25–77, respectively, and 45 years in the TT group, range 22–79 (*p* = 0.01). In the CRT group there were 167 patients (80%) with a tumour ≥6 cm, compared to 38 patients (59%) in the LND-CRT group and 20 patients (28%) in the TT group (*p* = 0.01). In the LND-CRT group there were more bulky nodes (80%, *n* = 53, *p* = 0.01) and more PELNM ≥ 2 cm (73%, *n* = 48, *p* < 0.01). The TT group included more stage IVA patients (15%, *n* = 14) and more PAOLNM (38%, *n* = 35) compared to the CRT group.

### 3.2. Treatment Characteristics

Differences in treatment characteristics between the groups were observed: 73% (*n* = 155) of the patients who underwent CRT received an additional RT boost, in contrast to 41% (*n* = 27) and 31% (*n* = 28) of the patients in the LND-CRT and TT group, respectively. EFRT was performed less frequently in the CRT group (32%, *n* = 67) compared to the LND-CRT group (49%, *n* = 32) and the TT group (52%, *n* = 47).

### 3.3. Oncological Outcome

Median follow-up of the three treatment groups was 49 months (range 2–208 months), during which 166 recurrences (45%) and 194 deaths (52%) were observed (seen in Table 2). In the CRT group 99 patients (47%) had a recurrence, compared to 37 patients (56%) in the LND-CRT group and 30 (33%) in the TT group, respectively (*p* = 0.02). Infield recurrences, shown in Table 2, were observed in 68 (41%) of 166 patients, which was similar across the three treatment groups (*p* = 0.17). The 5-year OS (Figure 1A) was comparable between the three treatment groups, with 53% (95% confidence interval 46–59%) in the CRT group, 45% (95% CI 33–56%) in the LND-CRT group and 53% (95% CI 40–64%) in the TT group, respectively (*p* = 0.46). In the adjusted analysis, 5-year OS was also comparable between the three treatment groups, as shown in Table 3. Similarly, 5-year DFS (Figure 1B) rates were comparable between treatment groups: 42% (95% CI 35–50%) in the CRT group, 38% (95% CI 25–50%) in the LND-CRT group and 53% (95% CI 41–62%) in the TT group, respectively (*p* = 0.11). After adjusting for confounders (seen in Table 3) no association was found between treatment and DFS (LND-CRT HR 1.17, 95% CI 0.80–1.71, TT HR 0.80, 95% CI 0.54–1.19).

### 3.4. Toxicity

Toxicities related to surgery and chemotherapy within the first 6 months after start of chemo- or radiotherapy are presented in Table 4. By definition, postoperative complications only occurred in the LND-CRT group (12%), of which infections were the most common (5%; *n* = 3). In the TT group there were two patients (2%) with grade 3 mucositis/stomatitis during chemotherapy, which was not reported in the CRT or LND-CRT groups (*p* = 0.05).

After adjusted analysis, there was no difference in chemotherapy related toxicity between treatment groups (Table 5).

## 4. Discussion

In this retrospective study, we investigated OS, DFS and toxicity in women with prognostically unfavourable LACC (defined as a tumour size of ≥6 cm, and/or PELNM of ≥2 cm and/or PAOLNM of ≥1 cm) who were treated with CRT, LND-CRT or TT. After adjusted survival analysis, no differences in OS or DFS between the treatment groups could be demonstrated. Additionally, no significant differences in treatment related toxicity could be identified between treatment groups.

The EMBRACE-I observational study has set the benchmark outcome for standard of care CRT with MRI-based image guided adaptive brachytherapy. Out of the 1341 patients included in the EMBRACE-I study, 98 patients had PAOLNM (suspicious nodes on CT, PET-CT or pathologically proven nodes after para-aortic LND) with a five-year OS of 61% with CRT and brachytherapy, whereas 190 patients with FIGO IIIB stage had a 5-year OS of 59% [17].

The results of the Dutch Deep Hyperthermia trial showed significantly more complete responses and a survival benefit for RT + HT compared to RT alone with the strongest effect in advanced cervical cancer [12]. A subgroup analysis of 114 patients with advanced cervical cancer was performed: Seventy percent of patients had FIGO stage IIIB, tumour diameter was ≥6 cm in 77% of patients and in 50 patients where nodal status was assessed, 35 patients (70%) were judged to have positive nodes [18]. This prompted further investigations into the added value of hyperthermia in LACC. More recently the outcomes of a randomized trial that included only stage III disease demonstrated a significant benefit of adding chemotherapy, i.e., 5-year OS after CRT 54% versus 46% after RT alone [19]. This initiated the RADCHOC trial, to investigate whether RT + HT should be preferred over CRT in bulky and/or FIGO stage ≥ III LACC with negative PAOLNM. Due to poor accrual (23%) and premature closing, the trial was underpowered for its primary outcome. The majority of the 84 included patients had tumours ≤6 cm and/or FIGO stage ≤ IIB. Still, the observed outcomes suggest that RT + HT provides an effective alternative treatment for CRT in LACC considering OS, pelvic recurrences and radiotherapy-related morbidity [13]. A meta-analysis with two included RCTs by Yea et al. concluded that CRT with HT significantly improves OS in LACC patients [14]. However, the RCT by Harima et al. included only 101 patients while excluding patients with PAOLNM [20]. The second RCT by Wang et al. reported a superior 5-year OS in the per protocol analysis, but not in the intention to treat analysis [21]. These results could be explained by the observation that more patients in the CRT group died from distant metastases and that there were about 5% more PELNM in the CRT group. Also, 97% of patients included had negative PAOLNM and 80% had negative PELNM. Additionally, the chemotherapy regimen used in the RCT by Wang et al. consisted of cisplatin 30 mg/m^2^ d1-3 and one cycle of 5-fluorouracil 350 mg/m^2^, d1-5. The thermal enhancement ratio is reported to be lower when 5-fluorouracil is used in an anticancer drug regimen and as such, the degree of the synergic effect of HT is unclear [22]. This is in contrast to our population with 95% positive PELNM and 35% positive PAOLNM and a NACT regimen consisting of platin-based chemotherapy with paclitaxel, which has a positive synergic effect [23]. As such, to our knowledge, this is the first survival analysis of a patient cohort with extensive nodal LACC treated with TT.

In line with our results, a retrospective study by Olthof et al. found no benefit in survival of lymph node debulking in patients with LACC and suspicious bulky nodes prior to CRT [8]. Marnitz et al. found that patients (*n* = 84) with microscopically positive versus negative PAOLNM had the same survival on surgical staging prior to CRT [24]. Leblanc et al. also reported that in 156 patients with locally advanced cervical cancer, resection of microscopically positive PAOLNM followed by EFRT and chemotherapy had the same survival as patients with pathologic negative lymph nodes who underwent pelvic irradiation and chemotherapy [25]. To facilitate the action of CRT by reducing tumor burden, Diaz-Feijoo et al. evaluated the feasibility and morbidity of performing pelvic LND associated with laparoscopic aortic LND in LACC. They concluded that indeed, it is feasible to perform laparoscopic pelvic LND in the same procedure as aortic LND in LACC without increasing surgical complications. Additionally, this procedure allows the clinician to know the lymph node status without delaying start of treatment with CRT [26]. The impact on survival, however, is not discussed.

Many studies have investigated the potential benefit of adjuvant chemotherapy. A recent meta-analysis found no benefit in unselected patients, despite the additional burden of toxicity [4]. The most recent randomized trial, OUTBACK, found no benefit in OS or DFS [27]. More recent trials investigate the role of immune checkpoint inhibition. The CALLA trial did not report a benefit of adding Durvalumab during and after CRT [28]. In contrast, the first interim analysis of the recently presented KEYNOTE-A18 study [29], with patients in node positive stage IB2-IIB or stage III-IVA disease irrespective of nodal status, found a 2-year DFS benefit of concurrent and adjuvant pembrolizumab. This RCT randomised patients 1:1 to receive either 5 cycles of pembrolizumab 200 mg every 3 weeks + CRT followed by 15 cycles of pembrolizumab 400 mg, or 5 cycles of placebo + CRT followed by 15 cycles of placebo. Twenty-four month DFS was 68% in the pembrolizumab + CRT group compared to 57% in the placebo + CRT group. In our cohort, 24-month DFS was 42% in the CRT group, 38% in the LND-CRT group and 53% in the TT group. More mature data are awaited from this trial, as well as from other ongoing trials investigating immune checkpoint inhibition.

Recently, results of the randomized phase III INTERLACE trial [5] were presented which will focus more treatment and research strategy towards NACT. In the INTERLACE trial 500 patients with squamous, adeno or adenosquamous carcinoma FIGO 2009 stage IB2, II, IIIB, IVA and stage IB1 node positive were recruited and randomised between CRT alone or NACT followed by CRT. The results show a significantly improved DFS and OS in LACC in the NACT + CRT arm, with 5-year DFS rate of 73% and 5-year OS of 80% in the NACT + CRT arm compared to 64% and 72% in the CRT only arm, respectively. Seventy-seven percent of patients had FIGO 2009 stage II disease and more than half of the patients (58%) had lymph node negative disease. To date, there are some results that suggest that this treatment strategy is feasible also in FIGO 2018 stage IIIC1 and stage IIIC2. The review by de Azevedo et al. reported one phase 2 trial of 22 patients with PELNM and/or PAOLNM with a median DFS and OS at 22 months of 68% and 81%, respectively [30]. Li et al. conducted a phase III trial comparing the efficacy and side effects of patients with FIGO 2018 stage IIB to IVA cervical cancer who were assigned to four cycles of NACT with cisplatin (40 mg/m^2^) and paclitaxel (60 mg/m^2^) weekly followed by CCRT or CCRT alone [31]. In a preliminary analysis of 50 patients, they reported after a median follow-up of 28 months, a 3-year OS rate of 84%, and a 3-year PFS rate of 74%. This analysis included 28 patients with FIGO 2018 stage IIIC1 and 6 patients with FIGO 2018 stage IIIC2. In our study population, 95% of patients had PELNM and 35% had PAOLNM, as we studied three heterogenous but real-life patient populations complicating cross-trial comparisons.

Several reasons exist as to why comparing treatment results between studies is complicated. First, there is no standard definition of bulky nodes in literature as definitions range from ≥1.0 to ≥2.4 cm [8]. Furthermore, studies on nodal boosting are likely to have included false positives, as suspicious nodes were not confirmed histologically and the positive predictive value of nodal imaging is only 55–96% [32]. This could positively affect survival rates. Furthermore, an analysis from the EMBRACE study reported large differences in stage distributions between FIGO stage 2009, FIGO stage 2018 and TNM stage as approximately 27% of patients had discrepant local tumour stages between clinical examination and MRI [33]. In the patients of the EMBRACE-I study 50% of the patients with FIGO 2009 IB1 were allocated to a higher stage on MRI, and 33% of those patients were allocated to IB2 due to large tumour size. Additionally, 31% of the patients with clinically staged IB1-IIa2 were upstaged to IIB with MRI. Such differences impact the overall management of these patients and consequently, survival.

In our cohort, after adjusted analysis, there was no difference in chemotherapy-related toxicity between treatment groups. Gao et al. reported on the acute and long-term toxicity in patients undergoing TT and concluded there was a high rate of chemotherapy-associated vascular toxicity and chronic kidney disease in the group treated with cisplatin [11]. This was in contrast to previous literature on toxicity in NACT and related to the higher cisplatin dose. In our study, mainly 70 mg/m^2^ was used, whereas in the EORTC study, a dose intensity of 25 mg/m^2^ per week was used and in the Gupta trial carboplatin was used [6,7]. In the study by Jakubowicz et al. the most frequent acute toxic effects in patients with LACC treated with concurrent CRT include haematological effects, nausea and vomiting, vulvo-vaginal disorders and urinary tract infections [34]. Similar rates of clinically relevant symptoms of acute haematological toxicity grade ≥ 3 occurred in the study by Gao et al. compared to Jakubowicz et al., i.e., 8% of patients and 7.5% of patients, respectively.

### 4.1. Strengths and Limitations

Important strengths of this study include the large number of subjects with prognostically unfavourable LACC treated with one of three treatment modalities and a long follow-up time. To our knowledge, it is the first time that outcomes of two alternative treatment modalities were compared with a conventionally-treated cohort.

Given the retrospective design, this study has multiple limitations. As the TT group from the Erasmus MC Hyperthermia database had been subject to a toxicity analysis, details of toxicity were most complete in the TT group. As such, fair comparison between toxicity data is not possible. All retrospective analyses are subject to underreporting of toxicity, especially in the CRT and the LND-CRT group.

This particular patient group is characterized by extensive locally and regionally advanced disease. Although our cohorts were relatively large, a case-matched control study was not able to overcome the restrictions of large intergroup variability. In the LND-CRT group, 72% of patients had PELNM ≥ 2 cm and 80% had one or more bulky nodes (PELNM or PAOLNM). We assume an important selection bias, as these patients would need to be fit for surgery. Furthermore, since there is no strict definition of bulky nodes and when to perform surgery, this could also influence the decision to perform LND or to treat with a RT boost [8]. Patients not fit for surgery would be more likely to receive CRT. Furthermore, there is another selection bias: we selected patients from the Erasmus MC Hyperthermia database and from the Dutch Cancer Society project ‘Chasing nodes, saving lives’ from the Netherlands Cancer Registry. Patients who initiated therapy but did progress under TT were not included, despite being treated in the same time period, and may have had the same tumour characteristics, although this is a limited number of patients (*n* = 6). We only evaluated chemotherapy-related toxicity and postoperative complications for the first 6 months and, as such, cannot assess the effect of long-term complications or the quality of life after treatment. Another limitation is that details of radiotherapy and brachytherapy dose and techniques were not included in this project. During the period 2009–2017, centers have transitioned to MRI image-guided adaptive brachytherapy, and combined intracavitary interstitial brachytherapy was introduced.

### 4.2. Future Research

Future studies should investigate the potential of carboplatin to reduce toxicity in triple therapy, newer techniques in radiotherapy and brachytherapy to achieve better local control with reduced side-effects and possible benefits of immune checkpoint inhibition. Ongoing trials to evaluate improvement in management beyond chemoradiotherapy in high-risk, locally advanced cervical cancer include, the phase II ATOMICC trial (NCT03833479), investigating dostarlimab and the phase III e-VOLVE Cervical Study (GOG-3092/ENGOT-cx19), studying volrustomig, a monovalent bispecific human IgG1 monoclonal antibody. Furthermore, neoadjuvant approaches may allow for extensive translational research opportunities and early evaluation of alternative endpoints such as response.

## 5. Conclusions

Patients with locally extensive advanced cervical cancer and bulky nodal involvement remain a prognostically unfavourable group. This retrospective study suggests that the OS, DFS and toxicity of triple therapy is similar to chemoradiotherapy with or without node-debulking.

## Figures and Tables

**Figure 1 cancers-16-00635-f001:**
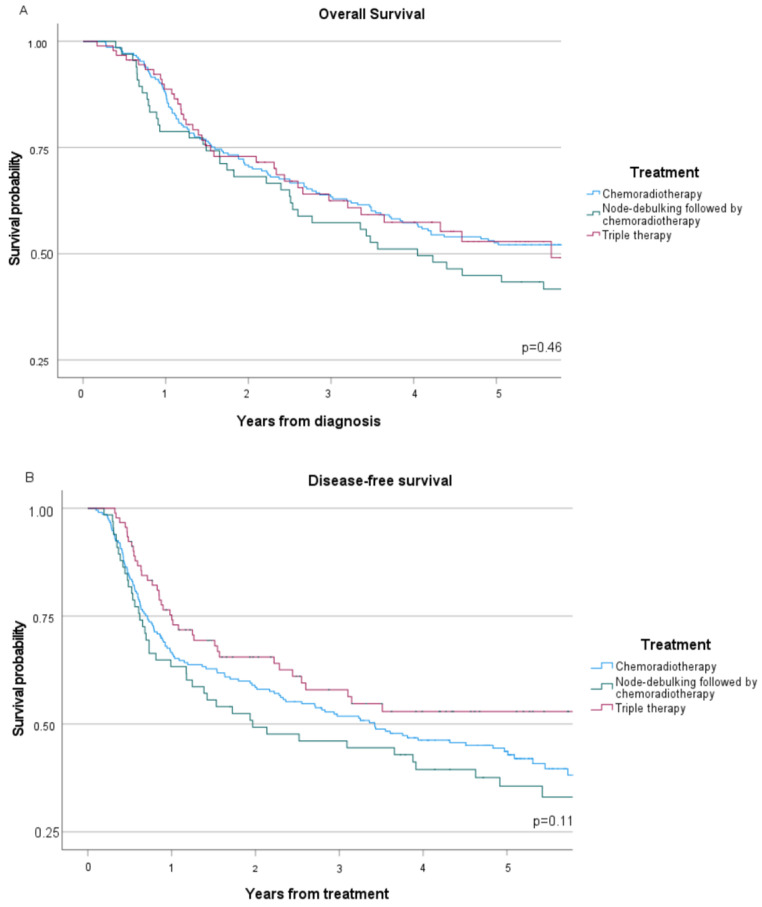
Kaplan—Meier functions for women with advanced stage cervical cancer, stratified by treatment strategy: (**A**) Overall survival; (**B**) Disease-free survival.

**Table 1 cancers-16-00635-t001:** Baseline characteristics per treatment group. CRT is the reference group.

Patient Characteristics	CRT (*n* = 213)	LND-CRT (*n* = 66)	TT (*n* = 91)	Missing (%)	*p*-Value
Median age, years (range)	50 (22–86)	50 (25–77)	45 (22–79)	-	0.01 *
BMI, kg/m^2^ (range)	25 (15–51)	25 (18–39)	24 (16–42)	23 (6)	1.00
Smoking, yes	74 (35)	25 (38)	41 (45)	49 (13)	0.02 *
FIGO 2009 stage				1 (0.3)	0.02 *
IB2	32 (15)	15 (23)	9 (10)		
IIA2	14 (7)	4 (6)	4 (4)		
IIB	90 (42)	28 (42)	46 (51)		
III	60 (28)	16 (24)	17 (19)		
IVA	17 (8)	3 (5)	14 (15)		
Histology				2 (1)	0.95
SCC	196 (92)	60 (91)	82 (90)		
Non-SCC	17 (8)	6 (9)	7 (8)		
Mean tumour size, mm (IQR)	64 (60–70)	60 (45–70)	59 (50–66)	9 (2)	0.01 *
Pelvic lymph node status on imaging				-	<0.01 *
Negative	3 (1)	1 (2)	13 (14)		
Lymph nodes 0.5–2.0 cm deemed positive	172 (81)	17 (26)	62 (68)		
Bulky ≥ 2 cm	38 (18)	48 (73)	16 (18)		
Para-aortic lymph node status on imaging				8 (2)	0.01 *
Negative	151 (71)	37 (56)	54 (59)		
Lymph nodes 0.5–1.0 cm deemed positive	19 (9)	13 (20)	21 (23)		
Bulky ≥ 1 cm	37 (17)	16 (24)	14 (15)		
**Treatment characteristics**					
Nodal boost				26 (7)	<0.01 *
Yes	155 (73)	27 (41)	28 (31)		
No	48 (23)	35 (53)	51 (56)		
Extended-field radiotherapy				13 (4)	0.01 *
Yes	67 (32)	32 (49)	47 (52)		
No	137 (64)	31 (47)	42 (46)		
Brachytherapy, yes	191 (90)	61 (92)	82 (90)	2 (1)	0.70
Time between diagnosis and start of chemo- or radiotherapy, median days § (IQR)	47 (38–58)	61 (49–77)	35 (20–49)	1 (0.3)	<0.01 *

Data are the number of patients with percentage in parentheses or median (range) or value (range); * statistically significant; CRT, chemoradiotherapy and brachytherapy only; LND-CRT, node-debulking followed by chemoradiotherapy and brachytherapy; TT, triple therapy consisting of neo-adjuvant chemotherapy followed by radiotherapy with concurrent hyperthermia and brachytherapy; FIGO, International Federation of Gynecology and Obstetrics; SCC, squamous cell carcinoma; § interval diagnosis and therapy does not include debulking procedure; IQR, interquartile range.

**Table 2 cancers-16-00635-t002:** Survival data across treatment groups.

Survival Data	CRT (*n* = 213)	LND-CRT (*n* = 66)	TT (*n* = 91)	Missing (%)	*p*-Value
Interval between start of chemo- or radiotherapy and death or recurrence, months	36 (1–146)	22 (2–129)	26 (4–198)	3 (1)	0.77
Recurrence, yes	99 (47)	37 (56)	30 (33)	1 (0.3)	0.02 *
Location of recurrence				7 (2)	0.11
No recurrence	114 (54)	29 (44)	60 (66)		
Central pelvic	15 (7)	6 (9)	13 (14)		
Lateral pelvic	22 (10)	10 (15)	16 (1)		
Para-aortic	23 (11)	12 (18)	12 (13)		
Distant metastases	72 (34)	27 (41)	16 (18)		
Infield recurrence, yes	33 (16)	15 (23)	20 (22)	9 (2)	0.17
Median time follow-up regarding overall survival, months	66 (3–169)	46 (5–171)	30 (2–208)	-	<0.01 *
Vital status, deaths	116 (55)	41 (62)	37 (48)	-	0.02 *

Abbreviations: * statistically significant; CRT, chemoradiotherapy and brachytherapy only; LND-CRT, node-debulking followed by chemoradiotherapy and brachytherapy; TT, triple therapy consisting of neo-adjuvant chemotherapy followed by radiotherapy with concurrent hyperthermia and brachytherapy; month, 30 days.

**Table 3 cancers-16-00635-t003:** Multivariable Cox Proportional Hazard analysis of progression and mortality.

	Overall Survival			Disease-Free Survival		
Variables	HR	95% CI	*p*-Value	HR	95% CI	*p*-Value
Therapy group						
CRT	1.00	Reference		1.00	Reference	
LND-CRT	1.40	0.95–2.06	0.09	1.17	0.80–1.71	0.43
TT	1.18	0.78–1.78	0.40	0.80	0.54–1.19	0.27
Age	1.03	1.02–1.04	<0.01 *	1.03	1.02–1.04	<0.01 *
FIGO 2009 stage						
<III	1.00	Reference		1.00	Reference	
≥III	1.63	1.19–2.24	<0.01 *	1.51	1.11–2.06	<0.01 *
Histology						
Squamous	1.00	Reference		1.00	Reference	
Non-squamous	1.33	0.81–2.18	0.26	1.24	0.75–2.04	0.40
Tumour size	1.00	1.00–1.01	0.50	1.00	1.00–1.01	0.52
Bulky nodes ‡						
Absent	1.00	Reference		1.00	Reference	
Present	1.04	0.73–1.48	0.84	1.16	0.82–1.64	0.41
Location suspicious node						
Absent	1.00	Reference		1.00	Reference	
Pelvic	1.54	0.55–4.31	0.41	1.45	0.52–4.03	0.48
Para-aortic	2.30	0.80–6.61	0.12	2.28	0.80–6.51	0.12

Abbreviations: * statistically significant; HR, hazard ratio; CI, confidence interval; CRT, chemoradiotherapy and brachytherapy only; LND-CRT, node-debulking followed by chemoradiotherapy and brachytherapy; TT, triple therapy consisting of neo-adjuvant chemotherapy followed by radiotherapy with concurrent hyperthermia and brachytherapy; FIGO, International Federation of Gynecology and Obstetrics; ‡ pelvic node ≥ 2 cm or para-aortic ≥ 1 cm.

**Table 4 cancers-16-00635-t004:** Complications and toxicities.

	CRT-O (*n* = 213)	LND-CRT (*n* = 66)	TT (*n* = 91)	*p*-Value
**Chemotherapy-related toxicity**	N (%)	N (%)	N (%)	
Nephrotoxicity	8 (4)	1 (2)	7 (8)	0.14
Ototoxicity	1 (1)	0	1 (1)	0.64
Mucositis/stomatitis	0	0	2 (2)	0.05
Neurotoxicity	1 (1)	0	3 (3)	0.06
Anaphylactic shock	0	0	0	N/A
Total	10	1	13	
Total patients †	7	1	8	0.38
**Postoperative complications**	N/A		N/A	
Intraoperative injury		2 (3)		
Infection		3 (5)		
Thromboembolism		1 (2)		
IC-admission		1 (2)		
Blood transfusion		1 (2)		
Bladder dysfunction		0		
Total		8		
Total patients †		6		
**Total**	22	17	17	
**Total patients †**	**19 (9%)**	**11 (17%)**	**13 (14%)**	**0.15**

Peri- and postoperative complications ≤ 30 days from surgery were scored by the Clavien—Dindo scale (grade 2 or higher). Chemotherapy-related toxicity ≤ 6 months after starting treatment was scored by the Common Terminology Criteria for Adverse Events version (CTCAE v4.03). Only grade 3 and higher were considered relevant and reported. Abbreviations: CRT, chemoradiotherapy and brachytherapy only; LND-CRT, node-debulking followed by chemoradiotherapy and brachytherapy; TT, triple therapy consisting of neo-adjuvant chemotherapy followed by radiotherapy with concurrent hyperthermia and brachytherapy; FIGO, International Federation of Gynecology and Obstetrics; † Some patients experienced multiple adverse events.

**Table 5 cancers-16-00635-t005:** Multivariable logistic regression analysis of patients.

	Toxicity		
Variables	OR	95% CI	*p*-Value
Therapy group			
CRT-O	1.00	Reference	
LND-CRT	2.04	0.92–4.55	0.08
TT	1.70	0.80–3.61	0.17
Age	1.02	0.99–1.04	0.14
Tumour size	1.01	0.99–1.03	0.23
FIGO 2009			
<III	1.00	Reference	
≥III	0.91	0.46–1.80	0.79
Location suspicious node			
Absent	1.00	Reference	
Pelvic	2.09	0.27–16.37	0.48
Para-aortic	2.31	0.29–18.69	0.43
Bulky node ‡			
Absent	1.00	Reference	
Present	1.84	0.97–3.52	0.06

Abbreviations: OR, odds ratio; 95% CI, 95% confidence interval; CRT, chemoradiotherapy and brachytherapy only; LND-CRT, node-debulking followed by chemoradiotherapy and brachytherapy; TT, triple therapy consisting of neo-adjuvant chemotherapy followed by radiotherapy with concurrent hyperthermia and brachytherapy; FIGO, International Federation of Gynecology and Obstetrics; ‡ pelvic node ≥ 2 cm or para-aortic ≥ 1 cm.

## Data Availability

The datasets generated and/or analyzed during the current study are available from the corresponding author on reasonable request.

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
