# Peer review of "Survival of Women with Advanced Stage Cervical Cancer: Neo-Adjuvant Chemotherapy Followed by Radiotherapy and Hyperthermia versus Chemoradiotherapy"

_cancers, 2024, doi:10.3390/cancers16030635_

Round 1
Reviewer 1 Report
Comments and Suggestions for Authors
The authors performed a retrospective cohort study to investigate overall survival, disease-free survival and toxicity of women who underwent either chemoradiotherapy with or without lymph node debulking (LND) or upfront chemotherapy followed by radiotherapy and hyperthermia (triple therapy). Although the study is interesting, there are some major concerns that need to be addressed:
1. As the authors mentioned in their discussion, there likely is substantial selection bias in this study. Although it might be difficult to overcome this due to the retrospective study design, the authors could describe in more detail the considerations for performing lymph node debulking prior to CRT in these patients, at least for those patients who were included in the 'Chasing nodes, saving lives' project.
2. Nodal status was assessed through imaging as per local protocol (i.e., computed tomography (CT), magnetic resonance imaging (MRI), or positron emission tomography (PET or PET-CT). Since PET imaging is known to be far more accurate than CT or MRI alone, information should be provided with respect to the number of patients who underwent PET/CT imaging for lymph node assessment.
3. In Table 1, the definition of positive pelvic lymph nodes is unclear. Was this determined on the basis of lymph node dimension as was done for paraaortic lymph node status? Please explain.
4. Toxicity results seem very underreported as only 16 and 3 patients out of the entire cohort reported nausea and fatigue, respectively, whereas these are the most common adverse effects of CRT. These data therefore seem to be unreliable. Authors should therefore consider to omit toxicity results..
Minor comments:
- Table 1: FIGO stage IIA should be changed to stage IIA2
- Page 4, line 173 'patients who' instead of 'patients that'
Reviewer 2 Report
Comments and Suggestions for Authors
This retrospective report examines CRT, lymph node volume reduction, and hyperthermic chemoradiation for cervical cancer.
Although a large number of cases are reviewed, there are issues with the study design, methods, and conclusions for publication.
There are various biases in the association between the treatment modalities and the cases studied, and drawing conclusions is difficult with the current methods of analysis.
It is unclear what you are trying to clarify in your objective; the benefit of lymph node volume reduction, or the combination of hyperthermia?
The methodology may need to be changed to a case-control study.
With regard to hyperthermia, an analysis of the temperature factor (thermal dose) should be included.
Reviewer 3 Report
Comments and Suggestions for Authors
The introduction and discussion are well written. The methodology and results need to be revised.
The calculation of the sample size is not explained
In Table 1, in columns 2, 3 and 4, the data in parentheses are not described
In table 1, in the second row of the last column, there is a significant number without an asterisk
In Table 1, it is not clear which columns have significant differences. That is, it is not specified in the post hoc test statistical method and is not shown in the table.
In the last line of the statistical analysis section, less than 0.05 is introduced as significant, but in Table 4, you consider 0.05 as significant, why?
One third of the resources used are old and can be replaced
Reviewer 4 Report
Comments and Suggestions for Authors
Dear Editor
The authors In the current study, evaluated overall survival, disease-free survival and treatment-related toxicities in patients with locally advanced cervical cancer treated with chemoradiotherapy with or without prior lymph node debulking or upfront chemotherapy followed by radiotherapy and hyperthermia
Although the study examined a sufficient number of patients n 370 retrospectively their conclusions are not different from those already known and present in the literature and the study lacks of novelty
The study concludes that Patients with locally extensive advanced cervical cancer and bulky nodal remain a prognostic unfavorable group. This retrospective study suggests that the OS, DFS and toxicity of triple therapy are similar to chemoradiotherapy with or without node-debulking
More information about FIGO molecular classification of patients and their possible correlation with variables could add novelty
Comments on the Quality of English Language
Dear Editor
The authors In the current study, evaluated overall survival, disease-free survival and treatment-related toxicities in patients with locally advanced cervical cancer treated with chemoradiotherapy with or without prior lymph node debulking or upfront chemotherapy followed by radiotherapy and hyperthermia
Although the study examined a sufficient number of patients n 370 retrospectively their conclusions are not different from those already known and present in the literature and the study lacks of novelty
The study concludes that Patients with locally extensive advanced cervical cancer and bulky nodal remain a prognostic unfavorable group. This retrospective study suggests that the OS, DFS and toxicity of triple therapy are similar to chemoradiotherapy with or without node-debulking
More information about FIGO molecular classification of patients and their possible correlation with variables could add novelty
Round 2
Reviewer 1 Report
Comments and Suggestions for Authors
Reviewer's suggestions have been adequately applied.
Author Response
Thank you for your kind and adequate suggestions.
Reviewer 2 Report
Comments and Suggestions for Authors
At present, unfortunately, I think the content is insufficient for publication. It suggests the existence of non-negligible biases in treatment methods and patient selection.
Author Response
Thank you for your time and effort to review our manuscript. Your suggestions and questions were relevant and we acknowledge that we are faced with multiple biases concerning the studied cases and treatment modalities.
- To further clarify our objective we added in the manuscript, page 1, abstract, line 33: to identify a potential role for triple therapy.
Reviewer 3 Report
Comments and Suggestions for Authors
I do not have any more suggestion. Thanks
Author Response
Thank you for your kind words and adequate suggestions.
Reviewer 4 Report
Comments and Suggestions for Authors
Dear Editor,
the authors addressed all my points I agree for publication
Comments on the Quality of English LanguageDear Editor,
the authors addressed all my points I agree for publication
Author Response

(The authors gave the same response as above.)
